# Geochemical and metagenomics study of a metal-rich, green-turquoise-coloured stream in the southern Swiss Alps

Antoine Buetti-Dinh[1,2☯]*, Michela Ruinelli[1☯], Dorota Czerski[3], Cristian Scapozza[3], Agathe Martignier[4], Samuele Roman[1,5], Annapaola Caminada[1], Mauro Tonolla[1,5,6]*

**1** Laboratory of Applied Microbiology (LMA), Department of Environment, Constructions and Design (DACD), University of Applied Sciences of Southern Switzerland (SUPSI), Bellinzona, Switzerland, **2** Swiss Institute of Bioinformatics, Lausanne, Switzerland, **3** Institute of Earth Sciences, University of Applied Sciences of Southern Switzerland (SUPSI), Trevano, Canobbio, Switzerland, **4** Department of Earth Sciences, University of Geneva, Geneva, Switzerland, **5** Alpine Biology Center Foundation, Bellinzona, Switzerland, **6** Microbiology Unit, Department of Botany and Plant Biology, University of Geneva, Geneva, Switzerland

☯ These authors contributed equally to this work.
* antoine.buetti@supsi.ch (ABD); mauro.tonolla@supsi.ch (MT)

**Data Availability Statement:** All Supporting information files are available from the Figshare database (accession number(s) DOI:10.6084/m9.figshare.12482660, 10.6084/m9.figshare.

## Abstract

The Swiss Alpine environments are poorly described from a microbiological perspective. Near the Greina plateau in the Camadra valley in Ticino (southern Swiss Alps), a green-turquoise-coloured water spring streams off the mountain cliffs. Geochemical profiling revealed naturally elevated concentrations of heavy metals such as copper, lithium, zinc and cadmium, which are highly unusual for the geomorphology of the region. Of particular interest, was the presence of a thick biofilm, that was revealed by microscopic analysis to be mainly composed of Cyanobacteria. A metagenome was further assembled to detail the genes found in this environment. A multitude of genes for resistance/tolerance to high heavy metal concentrations were indeed found, such as, various transport systems, and genes involved in the synthesis of extracellular polymeric substances (EPS). EPS have been evoked as a central component in photosynthetic environments rich in heavy metals, for their ability to drive the sequestration of toxic, positively-charged metal ions under high regimes of cyanobacteria-driven photosynthesis. The results of this study provide a geochemical and microbiological description of this unusual environment in the southern Swiss Alps, the role of cyanobacterial photosynthesis in metal resistance, and the potential role of such microbial community in bioremediation of metal-contaminated environments.

## Introduction

Microbial mats are composed of different horizontally stratified biofilms of microorganisms building a connected network with the ability to endure extreme environments such as hypersaline basins, sulphuretums, aquifers and sulfur springs, under prohibitive conditions for the growth of eukaryotic organisms [1, 2].

11967684, 10.6084/m9.figshare.13547438, 10.6084/m9.figshare.13523960, 10.6084/m9.figshare.11871444, PRJNA689378).

**Funding:** This study was funded through the canton of Ticino.

**Competing interests:** The authors have declared that no competing interests exist.

Sedimentary rock finds indicate a worldwide presence of microbial mats throughout the history of the Earth, as representative of first ecosystems together with stromatolites, and their role as modifiers of early atmosphere [3].

Modern biomats typically host a high biological diversity that includes bacterial, but also archaeal and eukaryotic communities [4]. This is partly due to dynamic physicochemical conditions that accommodate the needs of the different communities into interacting ecological niches [5–7], allowing them to carry out biological processes such as methanogenesis, denitrification, metal and sulfate reduction [8–10], as well as photosynthesis and nitrogen fixation in which Cyanobacteria play an important role [11]. Cyanobacteria are frequently found in diverse ecological niches including those harbouring high heavy metal concentrations [12]. Cyanobacteria's tolerance to such conditions is supposed to derive from their ability to synthesize extracellular polymeric substances (EPS). However, it is unclear how the oxidative stress induced by the combination of heavy metal ions present in polluted environments affects cyanobacterial physiology [13, 14]. Secreted EPS, membrane-bound or soluble, surround the cells in microbial communities through the formation of protective layers against oxidative and other sources of stress. In particular, their negative charge has been shown to play an important role in the protection against heavy metals-mediated oxidative stress, by depleting positively charged metal ions [15–17].

Therefore, microbial mats are of particular interest for studying microbial communities' diversity, structure and evolution, which contributes to their adaptation to extreme environments [18–20], and for their potential applications in bioremediation [4].

While biomats in different environments (for example hypersaline, acid, thermophillic, psychrophilic, oligotrophic, coastal mats) have been described previously [4, 7, 21–23], biomats in Alpine environments have been poorly studied so far. Here we investigate the microbial and chemical compositions of this biomat adapted to the Swiss Alpine environment (Fig 1). The peculiar chemical compositions of the biomat spring described in this study, principally composed of copper, cadmium and zinc, is unusual for the Swiss southern Alpine environment from a geological perspective [24, 25]. It is therefore of interest to investigate the formation of this biomat as a microbial adaptation to this unusual environment.

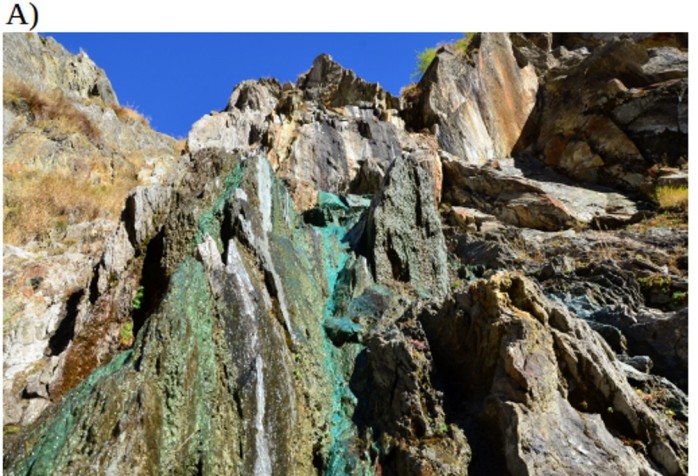 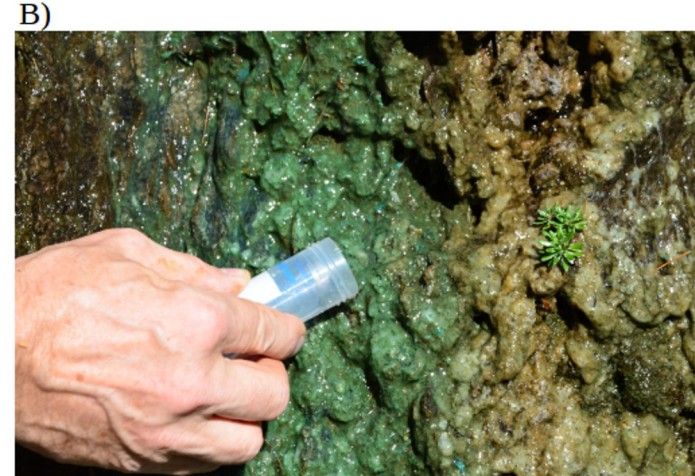

**Fig 1. Biomat spring in the Greina region, Camadra di Fuori / Sassina location in the Camadra valley.** This spring is well known in the region because of the green-turquoise colour of the biomat on the rocks [26]. The geographical position of the biomat spring is: CH1903+ / LV95 2'715'235, 1'160'895, at an elevation of 1'726 m asl. A) Overview of the biomat spring (summer 2015). B) Details of the sampling.

## Materials and methods

### Chemical profiling of the spring water

Water samples were taken using 500 ml disposable polypropylene bottles (Carl Roth, Arlesheim, Germany), they were kept at 4˚C and transported to the laboratory within 2 h.

The water chemical composition was measured using inductively coupled plasma mass spectrometry (ICP-MS), after the water samples were acidified using 0.3% HCl and 0.3% $NHO_3$. The conductivity was determined using a 5-ring conductivity measuring cell with cell constant $c = 0.7 \, cm^{-1}$ with integrated Pt1000 temperature sensor (Metrohm, art. 6.0915.100). The quantity of the anions (sulfate, fluoride, nitrate, chloride, nitrite and bromide) and cations (potassium and sodium) was measured using ion chromatography (IC, Metrohm, 850 Professional IC). Phosphate concentration was determined using UV-Vis colorimetric analysis (method SOP MSDA 628.1 [27]). Calcium and magnesium concentrations were measured using a calcium-selective electrode with polymer membrane (Metrohm, art. 6.0508.110). Strontium and zinc concentrations were measured with an inductively coupled plasma optical emission spectrometer (ICP-OES VISTA MPX Axial, Varian). Before analysis the samples were acidified with 1% $NHO_3$. All the other metal concentrations were measured using an inductively coupled plasma mass spectrometry (ICP-MS, iCAP-Q, ThermoScientific). Before analysis the samples were acidified with 0.3% HCl and 0.3% $NHO_3$.

### Microscopy

**Fluorescence microscopy.** The biofilm probes were examined by microscopy with a Zeiss Axiolab microscope in bright field and epifluorescence, using the F41 filter sets (AHF Analysentechnik HQ535/50, Q565LP and HQ610/75) for detection of phycoerythrin-containing autofluorescent cells [28].

**X-ray & scanning electron microscopy.** The rock samples were mounted on aluminum supports. They were covered with an ultra-thin coating of gold (10 nm) by low vacuum sputter, prior to imaging with a scanning electron microscope JEOL JSM 70001 FA (department of Earth Sciences, University of Geneva, Switzerland). Scanning electron microscope energy dispersive X-ray spectroscopy analyses (EDXS) were lead with a JEOL EX-94300S4L1Q detector. These analyses were acquired with an accelerating voltage of 15 kV, a beam current of 3.5 nA (acquisition times of 30 s). Gold (Au) is not taken into account in the semiquantitative quantification of the elements, as it is not part of the sample. Although not labelled, the characteristic energy peak of Au is visible on the spectrum at 2.12 KeV (S1 File).

### Sample collection and DNA extraction

The green-turquoise mucilage was collected during summer 2015 from the stream located in the Greina region, Camadra di Fuori / Sassina location in the Camadra valley (CH1903+ / LV95 2'715'235, 1'160'895, 1'726 m asl) using a falcon tube and stored at -20˚C. DNA was extracted from 10 g of mucilage using the DNeasy PowerMax Soil Kit (Qiagen) following the manufacturer protocol. After extraction, DNA was precipitated with ethanol and NaCl following the procedure suggested by the manufacturer. Finally, DNA was eluted in 50 $\mu$l of molecular grade $H_2O$. Quality and quantity of DNA was assessed spectrophotometrically using Nanodrop as well as the Quant-iT™ PicoGreen™ dsDNA Assay Kit (Invitrogen) combined with a TD700 Fluorometer (Turner Design) and the Qbit4 instrument (Thermofisher). In addition, the integrity of the DNA was checked by agarose gel electrophoresis.

## MinION sequencing

The metagenome of the green-turquoise mat was sequenced using a 1D ligation sequencing kit (SQK-LSK108). Sequencing was performed using an Oxford Nanopore Technologies (ONT) MinION flow cell R9.4 containing an initial number of 1'553 active nanopores for a duration run of 48 hours, using the MinKNOW software (v18.01.6). The ONT Guppy basecaller (v2.3.7) was further used to assign base names on the resulting chromatogram.

## Metagenome assembly and annotation

The fastq reads obtained by MinION sequencing were used to perform taxonomic classification using MetaMaps (v0.1) [29] against its "miniSeq+H" database (updated March 12[th] 2020) with BLAST NCBI taxonomy from Krona Tools (v2.7.1) [30]. The reads were also used to run Canu (v1.9) for assembling a metagenome with the following parameters: genomeSize = 5m, corOutCoverage = 10000, corMhapSensitivity = high, corMinCoverage = 0, correctedError-Rate = 0.105, redMemory = 32, oeaMemory = 32, batMemory = 200, maxMemory = 230G, nanopore-raw. Prokka (v13.1) was subsequently used to annotate the 3'421 contigs resulting from Canu assembly in conjunction with the NCBI BLAST's nt database (v2.10.0+). In addition, metagenomic binning was applied using the fastq reads and the assembled contigs longer than 10'000 nucleotides, using CONCOCT (v1.1.0) [31] with the "composition_file" option. Prokka was then used to annotate the contigs of the bins separately and CheckM (v1.1.2) [32] used to assess the completeness of the corresponding metagenome-assembled genomes (MAGs).

# Results and discussion

## Hydrochemical and geochemical analyses

The biomat spring is located at the contact between the migmatitic mica-alkali feldspar-plagioclase gneiss to the north and the biotite-muscovite-alkali feldspar gneiss to the south. These two units are the basement of Mels-, Röti- and Quarten-Formations (autochthonous cover of the Gotthard Massif deposited during the Triassic Period) composed by dolomitic/calcitic marble and cellular dolostone [24]. Just above the spring, a deposit of industrial minerals is inventoried in the Georesources information system of Switzerland [33]. This deposit belongs to the Triassic dolomitic marbles and contains, as elements, barium, fluorine, zinc, iron and copper, and, as minerals, barite, sphalerite, pyrite, chalcopyrite and fluorite, of hydrothermal origin [34].

Despite streaming through an area characterized by gneissic rocks with different mineralogical compositions, the hydrochemistry of the spring showed a calcium-sulfate type water strongly influenced by the presence of gypsum and carbonates in the Triassic rocks surrounding the area [24, 35]. In addition, the chemical profiling of the spring water indicated high concentrations of heavy metals such as aluminium ($\sim$1.25-fold higher than expected from granitic gneiss), barium ($\sim$2.6-fold higher than expected from dolomitic and calcitic marble and cellular dolostone) as well as lithium, manganese and strontium as expected from rocks of dolomitic origin, but also, surprisingly, it showed unusually high concentrations of copper (>60-fold), cadmium (>100-fold) and zinc (>650-fold) compared to the typical concentration range found in Alps (Table 1). The enrichment in the latter metals might be due to the presence of the hydrothermal veins previously mentioned, which contain minerals such as chalcopyrite and sphalerite [33, 35], and represents a significant difference with respect to concentrations found in typical Alpine environments, however being $\sim$100-fold lower than extreme acidophiles found in heavy-metal laden acid mine drainage waters [36, 37]. The

**Table 1. Chemical analysis of the spring water.**

| Chemical parameter | Measured value | Chemical parameter | Measured value |
|---|---|---|---|
| Conductivity at 20°C ($\mu$S/cm) | 539 | Phosphate (mg/L) | <0.01 |
| pH | 7.73 | Nitrite (mg/L) | <0.01 |
| Alcalinity at pH 4.3 (mmol/L) | 0.98 | Bromide (mg/L) | <0.01 |
| Sulfate (mg/L) | 254.5 | Molybdenum ($\mu$g/L) | 9.92 |
| Calcium (mg/L) | 116.4 | Uranium ($\mu$g/L) | 4.77 |
| Magnesium (mg/L) | 8 | Boron ($\mu$g/L) | 3.09 |
| Fluoride (mg/L) | 3.65 | Lead ($\mu$g/L) | 1.4 |
| Zinc (mg/L) | 3.39 | Selenium ($\mu$g/L) | 1.38 |
| Potassium (mg/L) | 2.8 | Antimony ($\mu$g/L) | 0.53 |
| Sodium (mg/L) | 1.2 | Cesium ($\mu$g/L) | 0.46 |
| Nitrate (mg/L) | 0.7 | Arsenic ($\mu$g/L) | 0.46 |
| Strontium ($\mu$g/L) | 574 | Cobalt ($\mu$g/L) | 0.34 |
| Copper ($\mu$g/L) | 305.14 | Tin ($\mu$g/L) | <0.20 |
| Chloride (mg/L) | 0.1 | Mercury ($\mu$g/L) | <0.20 |
| Ammonium (mg/L) | <0.1 | Bismuth ($\mu$g/L) | <0.20 |
| Aluminum ($\mu$g/L) | 40.19 | Vanadium ($\mu$g/L) | <0.1 |
| Manganese ($\mu$g/L) | 35.62 | Thallium ($\mu$g/L) | <0.1 |
| Cadmium ($\mu$g/L) | 30.86 | Silver ($\mu$g/L) | <0.1 |
| Barium ($\mu$g/L) | 25.32 | Iron ($\mu$g/L) | <0.1 |
| Nickel ($\mu$g/L) | 14.29 | Chromium ($\mu$g/L) | <0.1 |
| Lithium ($\mu$g/L) | 12.87 | Beryllium ($\mu$g/L) | <0.1 |

presence of a metarhyolite vein to the north of the study site could potentially explain the observed enrichment primarily of copper and zinc, but also of barium, cadmium, manganese and nickel.

The geochemical composition of the green-turquoise deposit collected at the spring was also investigated with the scanning electron microscopy analysis, and similar results were obtained (see S1 File). The analysis showed in fact typical elements contained in the crystal lattice of silicates of the gneissic rocks and elements as copper, zinc, iron and sulfur forming the minerals of hydrothermal origin.

## MinION metagenomics sequencing analysis

The sequencing generated a total of 5'874'348 reads with a mean length of 1'010 bp and a mean read quality of 9.7, which were further processed with the ONT Guppy basecaller resulting in 5'085'754 (86.6%) reads of quality score $Q > 7$. Taxonomic classification against the "miniSeq+H" index database (updated March 12th 2020) was used for evaluating biological diversity (Fig 2). The metagenome assembled contigs were annotated with Prokka before (S2 File) and after metagenomic binning (S3 File) in order to build different MAGs, further assessed for completeness, contamination and strain heterogeneity using CheckM (S4 File). These constituted the metagenomic data used to search for relevant genes and components involved in the biological functions described below.

**Overall bacterial diversity.** Proteobacteria were the most diverse group of bacteria present in the environment with a proportion of 58% (Fig 2). Among them, *Polaromonas spp.* of the Betaproteobacteria is known for being tolerant to elevated metal concentrations thanks to the metal-resistance genes for mercury, arsenate, chromate, and other heavy metals [38], and its role in pollutant degradation [39], as well as Rhizobiales (Alphaproteobacteria) such as

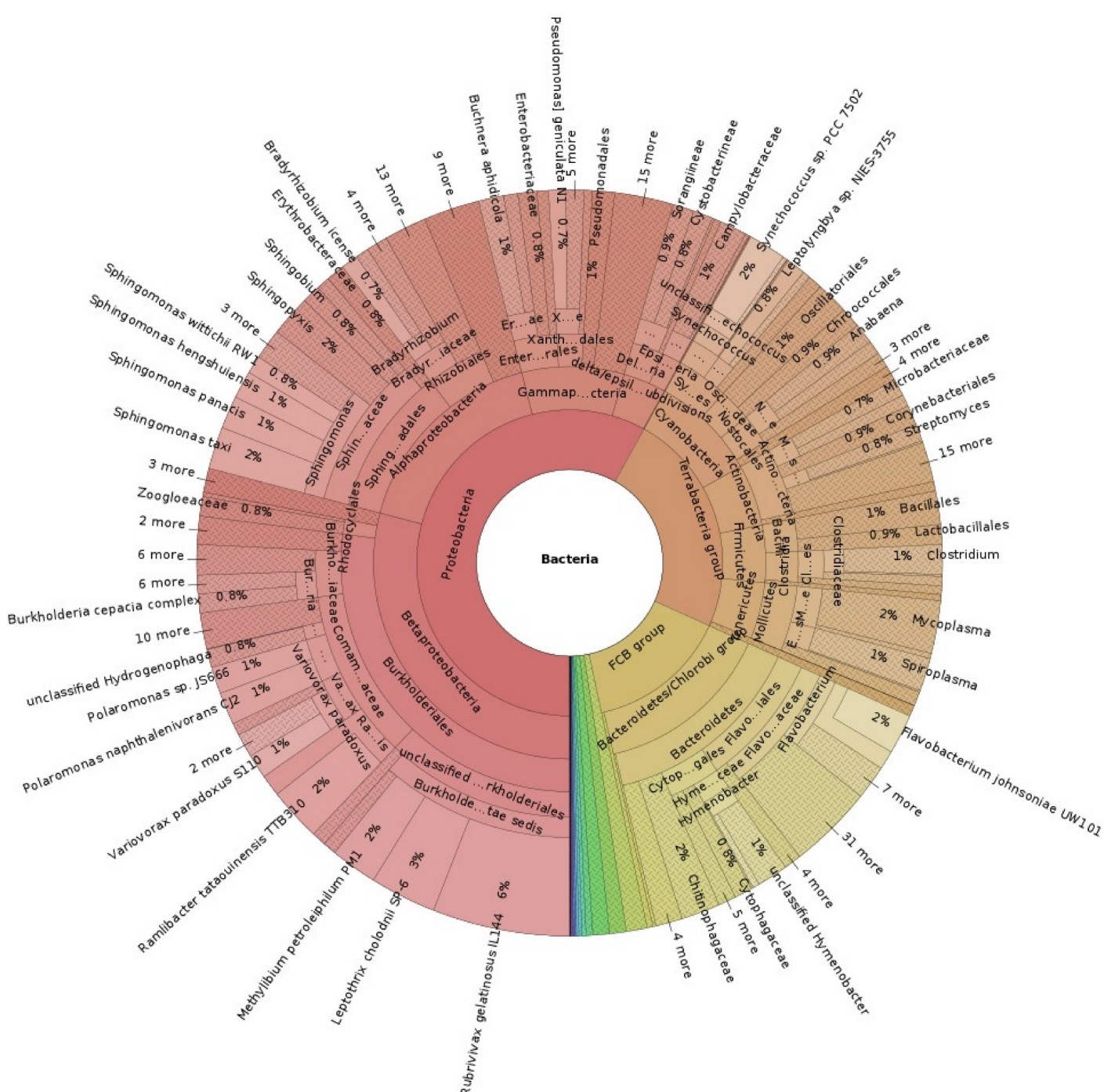

**Fig 2. Bacterial taxonomy based on MinION metagenomics sequencing, only reads representing more than 1% occurrence are represented (an interactive diagram is available in S5 File).**

*Rhodopseudomonas palustris* also involved in the removal of environmental pollutants by degrading chlorinated compounds [40].

Cytophaga, Bacteroidia, and Flavobacteriia are classes of the Bacteroidetes phylum that represented about 15% of the bacterial species and have been previously found in hypersaline mats [41] having a role in scavenging of Cyanobacteria biomass [42].

A)　　　　　　　　　　　　　　　　　　B)

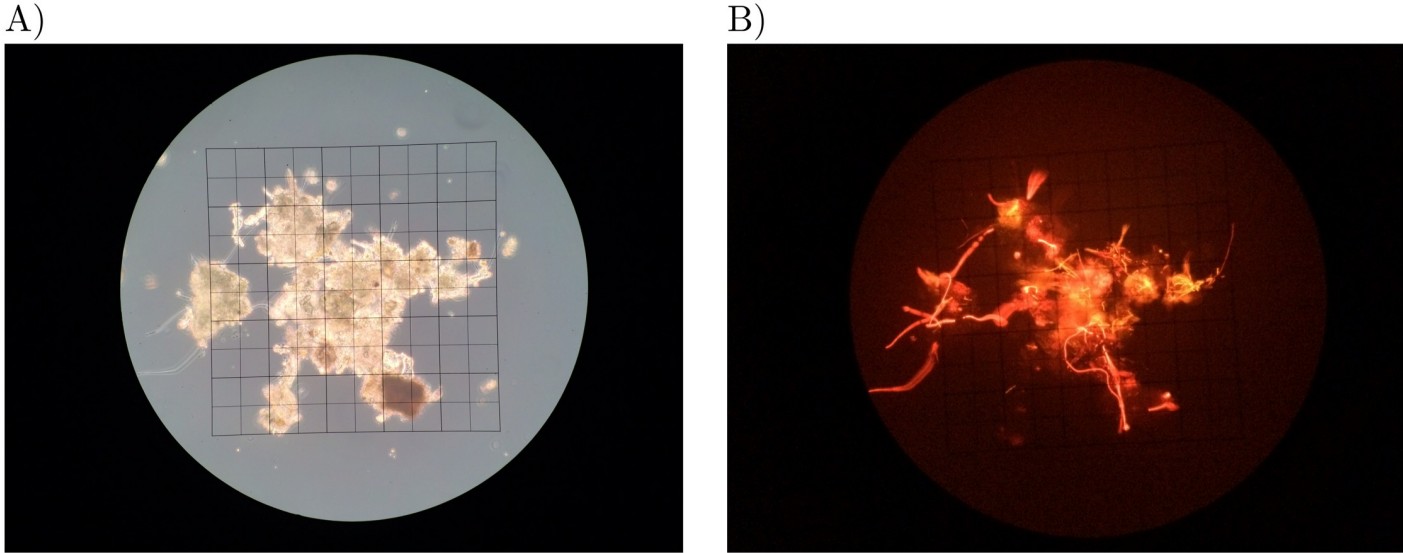

**Fig 3. Microscopy of biomat samples.** A) Biomat sample under 200x light microscopy magnification (grid side length = 635$\mu$m). B) Fluorescence microscopy at 552 nm of biomat's Cyanobacteria.

Terrabacteria represented 24% of the bacterial diversity, including Actinobacteria, Firmicutes, Tenericutes and Deinococci, known to be part of aquatic microbial biomat communities at low temperatures [43], aside of the most diverse group, *i.e.*, Cyanobacteria.

**Cyanobacteria diversity.** Fluorescence microscopy of the biofilm indicated the presence of Cyanobacteria (Fig 3) highlighted by the fluorescence emission range of the Cyanobacteria characteristic pigments.

Cyanobacteria diversity could be assessed based on MetaMaps analysis (Fig 4). Synechococcales were the most represented (42%), known to be involved in metal cycling in oceans' photic zone [44], followed by Oscillatoriophycideae (26%), such as *Gloeocapsa* and *Gloeothece spp.*, and Nostocales (25%), that are both considered as primary producers of phototrophic mats [4].

In addition, functional prediction of the genes based on the assembled metagenome indicated the presence of several genes involved in cyanobacterial metabolic activities. Cyanobacteria activity was suggested by cyanophycinase and cyanophycin synthetase genes, involved in the degradation and polymerization of Cyanobacteria-specific cyanophycin, respectively [45]. Diverse phycocyanobilin lyase subunits (CpcE,F,T,S) as well as phycocyanobilin:ferredoxin oxidoreductase (PcyA) and a putative phycocyanobilin lyase (CpcS) were also predicted by the metagenome annotation. The latter are involved in the light harvesting complexes [46], together with other photosynthetic antenna proteins (ApcA-E; CpcA,B,D,E-I,S,T; PetA-H,J,M; PsaA-F,I-M; PsbA-E,F,H-J,K,M,N,O,U,V,X,Y,Z), proteins involved in cyanobacterial oxidative phosphorylation (CyoE; NdhA-E; NdhH-N; Ppa; Ppk; SdhA,B,E) [47].

**Genes for nitrogen fixation.** Genes involved in nitrogen fixation were also found, such as the Nif-specific regulatory protein (NifA and NifS) and the nitrogen fixation protein (VnfA) found in Cyanobacteria (*Anabaena* genus) [48], as well as in bacteria, such as *Azotobacter* (Pseudomonadales) [49, 50], *Rhizobiales* [51, 52] and *Azospirillum* (Alphaproteobacteria) [53, 54], together with associated regulator proteins, such as a nitrogen regulatory protein P-II (GlnB) and a global nitrogen regulator (NtcA) commonly found in (cyano)bacteria, archaea and plants [55].

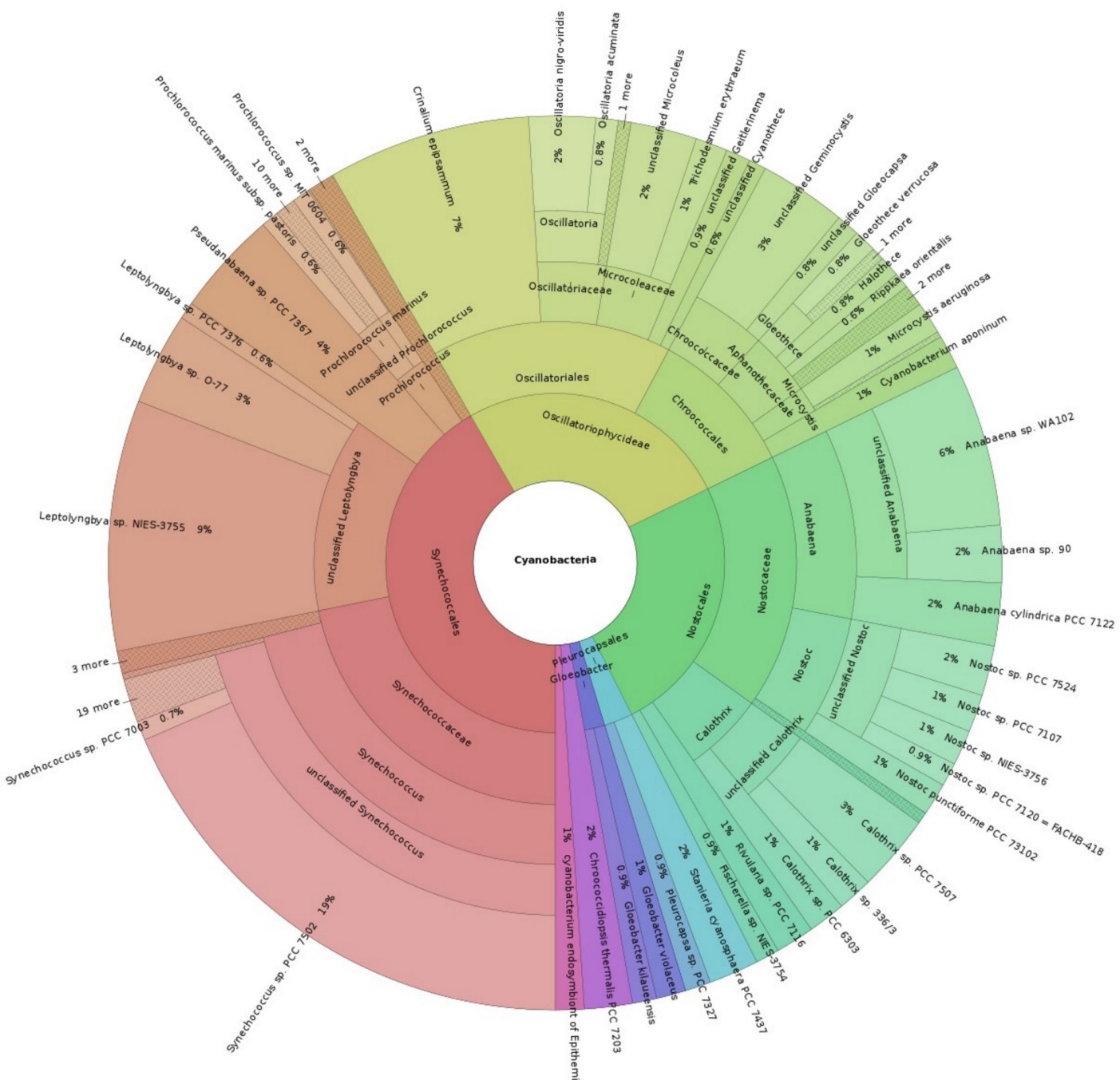

**Fig 4. Cyanobacterial taxonomy based on MinION metagenomics sequencing, only reads representing more than 1% occurrence are represented (an interactive diagram is available in S5 File).**

**Genes for microbial metal resistance.** Several genes involved in conferring tolerance/resistance to high metal concentrations were found in the annotated metagenome (S2 and S3 Files).

In particular, resistance against arsenic was represented by Acr3 and ArsA,C,H,M resistance effectors, which detect and stimulate the cellular response to arsenic [56, 57]. The genes

CzcA and B, CnrA and R, and CopA and B, were found in the annotated metagenome representing resistance mechanisms against elevated concentrations of cobalt, zinc, cadmium, nickel and copper [58–62], all of which were present in the sampled environment according to the chemical analysis (Table 1).

Further, genes coding for permeases related to the transport of iron were found (FeoB; FecA,E) described previously with the role of regulators of intracellular iron concentration [63–65], as well as PhnE,D and PstA,C for the uptake of phosphate at low extracellular concentrations [66] and for sulfate (CysT,W) [67], and sodium-translocating NADH-quinone reductase (subunits A,B,F).

Several components of other transport systems related to metals or chemical species measured in the environmental chemical analysis were found in the metagenomic data. Transporting ATPases for copper (ActP) [68], silver (SilP) [69, 70], cadmium/zinc/cobalt (CadA) [71, 72], zinc (ZiaA) [73, 74], calcium (PacL, YloB) [75, 76], magnesium (MgtA and MgtB) [77] and potassium (KdpA-C) [78], as well as other transporters were found for iron (FieF) [63, 79], ammonium (NrgA) [80, 81], magnesium (MgtE) [82], manganese (MntB,H,R) [83], nitrate/nitrite (NrtA and NrtP) [84], sodium (SdcS) [85], Zinc (ZitB) [86], cobalt/magnesium (CorA) [87, 88]. Import systems were found for phosphate (PstB, PhnD) [66, 89] and an antiporter for cadmium/cobalt/zinc *vs.* proton/potassium (CzcD) [62, 90, 91], as well as for sodium (NhaA,C,D, GerN, NhaS3) [92], molybdenum (ModA,B), nickel (LarO) or potassium (NhaP2) [93] *vs.* protons. Finally, genes coding for transcriptional regulators involved in sensing and uptake of phosphate (PhoB and PhoR) [94, 95] and zinc (Zur) [86] were identified.

**Genes involved in EPS synthesis.** The metagenome analysis revealed several genes predicted to be involved in EPS synthesis, which have been studied in various contexts from biosynthesis to biotechnological applications in bioremediation, for their capacity of heavy metal sorption [15, 16, 96–98]. Putative glycosyl-/acetyltransferases (EpsL, EpsM) and components of the Type II secretion system protein (EpsE, EpsF) were found in the annotated metagenome [99]. Also genes coding for glucans were found, such as 1,4-alpha-glucan branching enzyme (GlgB) [100], the 1,4-beta-D-glucan glucohydrolase (GghA) [101], 4-alpha-glucanotransferase (MalQ) [102], alpha-1,4-glucan:maltose-1-phosphate maltosyltransferase (GlgE) [103, 104], beta-glucanase (BglA) [105, 106], glucan synthase (NdvB) [107] and the Endoglucanase (Egl) [108]. Osmoregulated periplasmic glucans are part of EPS and have been found to play a role in bacteria that respond to harsh conditions such as osmotic [109] and heavy-metal stresses [110, 111]. The role of EPS in photosynthetic environments rich in metal has been demonstrated [17, 112–114] as the negative charge assumed in high pH, *i.e.*, during high photosynthetic regimes, might sequestrate toxic, positively charged metal ions.

## Potential and limitations

The metagenomics analysis approach benefits of modern sequencing tools, that encompass the sequencing itself made easier by user-friendly hardware such as the MinION sequencer, together with the accompanying nucleic acid extraction kits, as well as software for data analysis such as MetaMaps that take advantage of long reads for optimizing information retrieval from public databases. A limitation of metagenomic studies is that they rely on the genomic DNA and therefore only allow to infer the genetic composition of the organisms living in a studied environment, without providing evidence for actual activity of the genes that are identified, for which further analysis would be required for the assessment of gene expression, such as metatranscriptomics. However, the validity of our metagenome-based approach is supported by our findings that revealed a multitude of genes conferring tolerance to the chemicals

we detected by chemical profiling the environment, as they were likely retained through natural selection.

## Conclusions

In this study we described an environment in the southern Swiss Alps which is atypical for the region from its peculiar chemical composition. We characterized this environment from a geochemical and microbiological perspective.

Geochemical analysis revealed a complex rock composition due to the merging of different geological units during rock formation of the surrounding area. This determines the unusual chemical composition of the outflowing water, characterized by high concentrations of heavy metals. Interestingly, microbiological analyses based on microscopy and metagenomics revealed an ecosystem adapted to these conditions. The ecosystems appeared as a green-turquoise biomat mainly composed of Proteobacteria, Bacteroidetes, and Cyanobacteria, along with other taxonomic groups. In particular, the substantial presence of Cyanobacteria in the biomat underlines its photosynthetic activity, where Cyanobacteria play a crucial role of primary producers and nitrogen fixators, thereby providing the underlying building blocks for the complex nutrient flow network involving the other microorganisms composing the biomat. The photosynthetic activity in such environments was suggested to improve tolerance to high heavy metal concentrations, by an increased pH that drives metal cations sequestration by EPS. Indeed, genes for synthesis of EPS were found in the assembled biomat metagenome, along with several other genes involved in diverse mechanisms of metal resistance. Altogether, this study allowed the first characterization of this unusual Swiss Alpine ecosystem from a geochemical and microbiological perspective. Several open questions remain, for example regarding the ecosystem's seasonal dynamics, as well as potential applications to adapt such microbial community for bioremediation of anthropogenically contaminated environments rich in heavy metals.

## Supporting information

**S1 File. Scanning electron microscopy analysis.**
(TXT)

**S2 File. Annotated metagenome.**
(TXT)

**S3 File. Annotated MAGs.**
(TXT)

**S4 File. CheckM evaluation of MAGs.**
(TXT)

**S5 File. Interactive Krona taxonomic representation diagram.**
(TXT)

**S1 Raw data. SRA/BioProject accession number: PRJNA689378.**
(TXT)

## Author Contributions

**Conceptualization:** Cristian Scapozza, Mauro Tonolla.

**Data curation:** Antoine Buetti-Dinh.

**Formal analysis:** Antoine Buetti-Dinh, Michela Ruinelli, Dorota Czerski, Cristian Scapozza, Agathe Martignier, Samuele Roman, Annapaola Caminada.

**Methodology:** Antoine Buetti-Dinh, Michela Ruinelli.

**Project administration:** Mauro Tonolla.

**Software:** Antoine Buetti-Dinh.

**Validation:** Antoine Buetti-Dinh.

**Visualization:** Antoine Buetti-Dinh.

**Writing – original draft:** Antoine Buetti-Dinh, Michela Ruinelli, Cristian Scapozza, Mauro Tonolla.

**Writing – review & editing:** Antoine Buetti-Dinh, Mauro Tonolla.

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
