## [Decision Letter · Decision Letter 0]

2 Oct 2020

PONE-D-20-23974

Geochemical and metagenomics study of a metal-rich, green-turquoise-coloured stream in the southern Swiss Alps

PLOS ONE

Dear Dr. Buetti-Dinh,

Thank you for submitting your manuscript to PLOS ONE. After careful consideration, we feel that it has merit but does not fully meet PLOS ONE’s publication criteria as it currently stands. Therefore, we invite you to submit a revised version of the manuscript that addresses the points raised during the review process.

We look forward to receiving your revised manuscript.

Kind regards,

Erika Kothe

Academic Editor

PLOS ONE

Journal Requirements:

3. Please upload a copy of Supporting Information S1 File, S2 File and S3 File which you refer to in your text on page 8.

Additional Editor Comments:

The reviewer has made very good comments. Please carefully check the phrasing to avoid over-interpretation. As the absolute must-do in your case is the deposition of the data in a public library. Since that may some time, I would decide on major revision (to give you time), although from the content of the expert reviewer's comments it would qualify for minor revision.

Reviewers' comments:

Reviewer's Responses to Questions

**Comments to the Author**

1. Is the manuscript technically sound, and do the data support the conclusions?

Reviewer #1: Yes

2. Has the statistical analysis been performed appropriately and rigorously? 

Reviewer #1: Yes

3. Have the authors made all data underlying the findings in their manuscript fully available?

Reviewer #1: No

4. Is the manuscript presented in an intelligible fashion and written in standard English?

Reviewer #1: Yes

5. Review Comments to the Author

Reviewer #1: This manuscript presents a metagenome study of metal-rich aquatic environment in the Swiss alps. What a terrific field site!

This is a great focused study using analytical profiling of the water as well as metagenome study of the microbial mats to understand community makeup.

Abstract:

"genes that have been selected to allow microbial adaptation" ->

Clarify why you are surprised to find EPS genes, aren't these known to be encoded in Cyanobacteria genomes?

some of the language is a little imprecise and could be helpful to improve.

Line 2: "are composed of different biofilms of microorganisms" - could you be more clear is there one biofilm comprised of many organisms, or are there layers of films each heterogeneous compositions of microbes?

Methods:

- please provide version numbers for the tools used (MetaMaps, Krona Tools) some information about what version the miniSeq+H database was searched to assign taxonomic names?

- not clear why pacbio-raw was used when running ONT reads, isn't -nanopore-raw is an option?

- the canu option listed 5m - so is that reasonable genome size for a metagenome? I suppose it is just an estimate to get depth of coverage correct for how it runs- but you might evaluate after the assembly whether contigs have different depth of coverage values indicating organisms in different abundances in the sample.

- what is "the blast database" indicated on line 101?

- I'm surprised no metagenome binning applied to better adjust for individual species genomes?

line 230:

"that take profit of long reads" -> this phrasing could be "profit from" or "take advantage of"

line 233:

"potential gene program", the term "gene program" is a little confusing - but just may depend on how you want to word it.

It seems helpful to spend a little more time contrasting the levels of chemicals found with those in other aquatic environments to clarify how extreme this makeup is? The results/discussion do not cover extensively an interpretation of the quantitative values found - are they at extremes of most life? are they beyond what is found in most streams?

The authors' argument that the microbes in the community have adapted by selection to the environment are attractive but lack much statistical rigor. Just counting up genes without contrasting to an alternative model isn't sufficient. For example - if you focused on one of the most well assembled microbes (again a benefit to binning the contigs to species so you can assess the overall gene content of one of the Metagenome-Assembled Genome (MAG) - you could contrast the copy number of metal resistance implicated genes or transporter genes with the gene set found in a sister lineage which was not from non-extreme conditions.

- if you do this, then some assessment of completeness of the genomes - eg BUSCO or CheckM scores.

- annotation of each of these individually may provide slight better results if gene predictors were able to run and train on each genome set individually. To that end I am not sure if Prokka would perform better on the annotation if the data were binned and run each one at a time.

Were there any evidence of archaea or non-bacteria in your metagenomes?

Just to comment on this - I believe there may be signal towards understanding if adaptation has occurred but that would be better addressed with something quantifiable - eg accelerated rates of molecular evolution; expansions of copy number of gene families that underlie EPS or metal tolerance.

Data availability

The metagenome raw data and annotated assembly much be deposited in the INSDC public sequence archive (genBank, EMBL, DDBJ). Supl file 3 is not a substitute for depositing in a sequence archive. Likewise the raw fast5 data from ONT need to be deposited into SRA and a BioProject and SRA project ID assigned to the unassembled dataset.

6. PLOS authors have the option to publish the peer review history of their article (what does this mean?). If published, this will include your full peer review and any attached files.

Reviewer #1: **Yes: **Jason Stajich

---

## [Author Response · Author response to Decision Letter 0]

30 Jan 2021

Please see the file uploaded with the response to Reviewer comments ("Response_to_Reviewers.pdf").

---

## [Decision Letter · Decision Letter 1]

8 Mar 2021

Geochemical and metagenomics study of a metal-rich, green-turquoise-coloured stream in the southern Swiss Alps

PONE-D-20-23974R1

Dear Dr. Buetti-Dinh,

We’re pleased to inform you that your manuscript has been judged scientifically suitable for publication and will be formally accepted for publication once it meets all outstanding technical requirements.

Kind regards,

Erika Kothe

Academic Editor

PLOS ONE

Additional Editor Comments (optional):

Reviewers' comments:

Reviewer's Responses to Questions

**Comments to the Author**

1. If the authors have adequately addressed your comments raised in a previous round of review and you feel that this manuscript is now acceptable for publication, you may indicate that here to bypass the “Comments to the Author” section, enter your conflict of interest statement in the “Confidential to Editor” section, and submit your "Accept" recommendation.

Reviewer #1: All comments have been addressed

2. Is the manuscript technically sound, and do the data support the conclusions?

Reviewer #1: Yes

3. Has the statistical analysis been performed appropriately and rigorously? 

Reviewer #1: Yes

4. Have the authors made all data underlying the findings in their manuscript fully available?

Reviewer #1: Yes

5. Is the manuscript presented in an intelligible fashion and written in standard English?

Reviewer #1: (No Response)

6. Review Comments to the Author

Reviewer #1: These revisions strengthen the paper and I am supportive that this revision is suitable for publication. The authors have addressed the comments and I am happy to see this work published and shared broadly.

7. PLOS authors have the option to publish the peer review history of their article (what does this mean?). If published, this will include your full peer review and any attached files.

Reviewer #1: **Yes: **Jason Stajich

---

## [Editor Report · Acceptance letter]

9 Mar 2021

PONE-D-20-23974R1 

Geochemical and metagenomics study of a metal-rich, green-turquoise-coloured stream in the southern Swiss Alps 

Dear Dr. Buetti-Dinh:

I'm pleased to inform you that your manuscript has been deemed suitable for publication in PLOS ONE. Congratulations! Your manuscript is now with our production department. 

Kind regards, 

on behalf of

Prof. Dr. Erika Kothe 

Academic Editor

PLOS ONE